# Tokenization Consistency Matters for Generative Models on Extractive NLP Tasks

**Kaiser Sun**♣* **Peng Qi**♠† **Yuhao Zhang**♠†
**Lan Liu**♠ **William Yang Wang**♠ **Zhiheng Huang**♠
♣Paul G. Allen School of Computer Science & Engineering, University of Washington
♠AWS AI Labs
huikas@cs.washington.edu {pengqi,yhzhang}@amazon.com
{liuall,wyw,zhiheng}@amazon.com

## Abstract

Generative models have been widely applied to solve extractive tasks, where parts of the input is extracted to form the desired output, and achieved significant success. For example, in extractive question answering (QA), generative models have constantly yielded state-of-the-art results. In this work, we study the issue of tokenization inconsistency that is commonly neglected in training these models. This issue damages the extractive nature of these tasks after the input and output are tokenized inconsistently by the tokenizer, and thus leads to performance drop as well as hallucination. We propose a simple yet effective fix to this issue and conduct a case study on extractive QA. We show that, with consistent tokenization, the model performs better in both in-domain and out-of-domain datasets, with a notable average of +1.7 $F_1$ gain when a BART model is trained on SQuAD and evaluated on 8 QA datasets. Further, the model converges faster, and becomes less likely to generate out-of-context answers. Our results demonstrate the need for increased scrutiny regarding how tokenization is done in extractive tasks and the benefits of consistent tokenization during training.[1]

## 1 Introduction

Pretrained sequence-to-sequence (seq2seq) models have achieved remarkable success in a wide range of tasks (Lewis et al., 2020, Raffel et al., 2020). As an important component of the models, tokenizer is frequently discussed, including different tokenization methods and the model's robustness to different tokenization (Provilkov et al., 2020).

In our work, we identify an issue of tokenization consistency that affects the performance of seq2seq models. Specifically, when a seq2seq task has extractive nature, i.e., parts of the output text

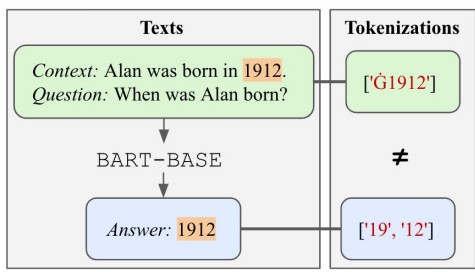

Figure 1: An example of tokenization inconsistency in training a BART model (which uses the BPE tokenizer) for extractive QA. The number "1912" is tokenized differently alone (blue) and in context (green), because unlike in context, the answer is often provided without preceding spaces, which triggers different BPE merging rules during tokenization. We propose to extract the tokenized answer from context (green) for training.

are extracted from the input text, the desired output could be tokenized differently from how it is tokenized in the input, leading to *tokenization inconsistency* during model training. For example, in extractive question answering (QA), which takes a context-question pair as input and outputs a span of context as answer, the answer might be tokenized differently from the span in the context (Figure 1).

Different variants of inconsistent tokenization have been found in several previous works in other tasks such as semantic parsing, open-domain question answering (Petrak et al., 2022; Rosenbaum et al., 2022; Yu et al., 2023), but important issues like this are often deemed not analysis-worthy, and no universal solution nor in-depth analysis have been presented. This seemingly minor difference in tokenization alters referential integrity and can result in a notable impact on model performance during inference. We use extractive QA as a case study and propose a simple and effective approach to mitigate this issue – extracting the tokenized answer from the context for training. We discover that, when fine-tuning with consistently tokenized instances, the model 1) achieves better in-domain and

---

*Work done during an internship at AWS AI Labs.
†Equal contribution.

[1]Code is available at https://github.com/KaiserWhoLearns/ConsistentTokenization.

| Dataset | BART | T5 | Dataset | BART | T5 |
|---------|------|-----|----------|------|------|
| SQuAD | 96.1 | 5.0 | BioASQ | 83.6 | 25.3 |
| TriviaQA | 95.1 | 72.8 | TextbookQA | 96.9 | 21.8 |
| NewsQA | 85.0 | 0.0 | DuoRC | 85.3 | 27.5 |
| NQ | 99.6 | 0.4 | SearchQA | 68.2 | 14.7 |

Table 1: Percentage of training instances whose tokenized gold answers do not exist verbatim in the tokenized context using BART and T5 tokenizer.

out-of-domain performance (+0.9% $F_1$ in-domain and +2.0 % zero-shot out-of-domain), 2) converges faster during training, and 3) is less likely to generate out-of-context output (i.e. less likely to hallucinate textually). With our findings, we would like to note that inconsistent tokenization can affect any tasks that can be cast as seq2seq tasks with an extractive nature beyond QA and call on researchers and practitioners to consider applying the consistent tokenization technique during training.

## 2 Related Work

Byte-Pair-Encoding (BPE) (Sennrich et al., 2016), language-model-based segmentation (Kudo, 2018), and their variants (Kudo and Richardson, 2018) are commonly used as the tokenizers for NLP models due to their simplicity and universality.

Recent research has identified these tokenization approaches as sources of poor model generalization. For example, BPE has been shown to produce less ideal morphological segmentation (Bostrom and Durrett, 2020; Provilkov et al., 2020), and the same text can be tokenized differently when different BPE rules are applied (Kudo, 2018). Existing approaches either modify model training by using stochastic inputs or by modifying how BPE segmentation is done to improve model robustness and generalization (Provilkov et al., 2020; He et al., 2020; Vilar and Federico, 2021, *inter alia*). In contrast, we propose a simple and deterministic approach that does not rely on altering the segmentation approach in any way, so that the original training strategy can be preserved.

Besides generic approaches, researchers have also investigated domain-specific approaches to improve segmentation on specific texts (Geva et al., 2020; Petrak et al., 2022; Rosenbaum et al., 2022). Geva et al. (2020) and Petrak et al. (2022) propose approaches that aim at overcoming the inconsistency of tokenizing numbers and enhancing the model's numerical reasoning ability. Rosenbaum et al. (2022) use "space-joined tokens" to resolve

many string-matching anomalies after tokenization that lead to unfair evaluation in semantic parsing. We instead look at inconsistency from a more general perspective by making our method applicable to any tasks with an extractive nature.

In addition to subword segmentation methods, another line of research focuses on character- and byte-level modeling as procedures that are free of tokens (Graves, 2013; Al-Rfou et al., 2019; Xue et al., 2022; Tay et al., 2022). While tokenization-free methods show potential to surpass subword segmentation approaches, we remain focused on the consistency issue of subword tokenizations, as most prevalent state-of-the-art models rely on subword tokenizations (Chung et al., 2022; Touvron et al., 2023).

## 3 Consistent Tokenization: What It Is and How To Achieve It

**Consistent Tokenization** Consider a seq2seq task, which takes text $x = (x_1, ..., x_n)$ as input, and outputs $y = (y_1, ..., y_m)$. When the task is extractive, there exists two sets of indices, $\mathcal{I}$ and $\mathcal{J}$, such that $x_{\mathcal{I}} = y_{\mathcal{J}}$. In the example of Figure 1, $x$ is the context-question pair, $y = 1912$, and $x_{\mathcal{I}} = y_{\mathcal{J}} = 1912$.

Let tokenization be a function $T$ that maps text into a sequences of token ids. Suppose $x_{\mathcal{I}} \mapsto T(x)_{\mathcal{I}'}$ and $y_{\mathcal{J}} \mapsto T(y)_{\mathcal{J}'}$. Here, $\mathcal{I}'$ denotes the set of indices that maps to the $x_{\mathcal{I}}$ in the tokenized input, while $\mathcal{J}'$ denotes the position of $y_{\mathcal{J}}$ in the tokenized output. Note that $T(x)_{\mathcal{I}'} = T(y)_{\mathcal{J}'}$ is not always true: in Figure 1, $x_{\mathcal{I}}$ is mapped to the ids of "Ġ1912", while $y_{\mathcal{J}}$ is mapped to the ids of "19" and "12", because there is no preceding space in $y_{\mathcal{J}}$. We call it *inconsistent tokenization* when $T(x)_{\mathcal{I}'} \neq T(y)_{\mathcal{J}'}$. Analogously, tokenization is *consistent* when $T(x)_{\mathcal{I}'} = T(y)_{\mathcal{J}'}$. Inconsistent tokenization could emerge due to the existence of preceding space, numbers, or punctuation when using the vanilla BPE tokenizer. SentencePiece tokenizers are less subject to inconsistency because one of the underlying reasons, preceding space, is mitigated by the combined implementation of vanilla BPE and unigram.

**Consistent Tokenization Training** When the input and output are tokenized inconsistently, the task can no longer be solved by simply extracting the output from the input token ids, but requires an additional step from the model to "paraphrase" the input ids into the output token ids that do not exist

in the input ids. Therefore, learning with inconsistently tokenized instances becomes an inherent predicament for model compared to learning with their consistently tokenized counterparts.

Instead of tokenizing output *in situ*, we propose to retrieve $T(y)_{\mathcal{J}'}$ from $T(x)_{\mathcal{I}'}$ such that the tokenization is always consistent among all $x$, $y$ pairs. Compared to proposing a new tokenization method that is immunized to inconsistency, this is a simple yet effective fix that every researcher can implement without any non-trivial effort and without the need to pretrain the model again.

## 4 Case Study: Extractive QA with Generative Models

In this work, we use extractive QA as a representative for extractive tasks. Extractive QA is an ideal candidate for studying the effect of consistent tokenization, since its output is always a substring of the input. Recent work has demonstrated that applying generative models to this task leads to great performance gains (Lewis et al., 2020; Raffel et al., 2020; Brown et al., 2020; Izacard and Grave, 2021) and greater potential to unify different input formats (Khashabi et al., 2020).

### 4.1 Task Description

In extractive QA, the model is given a question with a context, and expected to output a span (substring) of the context as an answer. Extractive models are typically configured with the question and context as the input, and trained to return start and end indices to indicate the location of the predicted answer in the context. For generative models, index prediction task is replaced with a task of directly predicting the answer string from the decoder of a seq2seq model, thus the need to tokenize the answer string separately from the context.

### 4.2 Experimental Setup

**Data** MRQA (Fisch et al., 2019) is used as benchmark for the experiments. We first fine-tune the model on one of the datasets among SQuAD (Rajpurkar et al., 2016), TriviaQA (Joshi et al., 2017), and NewsQA (Trischler et al., 2017). Then, we evaluate the model on its in-domain test set and seven out-of-domain test sets (SQuAD, TriviaQA, NewsQA, NaturalQuestions (NQ; Kwiatkowski et al., 2019), BioASQ (Tsatsaronis et al., 2015), TextbookQA (Kembhavi et al., 2017), DuoRC (Saha et al., 2018), SearchQA (Dunn et al., 2017)).

All the datasets are in MRQA format. We show the percentage of inconsistently tokenized instances in each dataset in Table 1. We also find that the issue is still prevalent even when the most common and easily resolvable source of inconsistency, prefix space, is addressed, suggesting the remaining to be a non-trivial issue (Appendix A).

**Model Choice** As one of the widely used generative models, BART-base (Lewis et al., 2020) is used for experiments. Compare to T5, which uses SentencePiece (Kudo and Richardson, 2018) as tokenizer, BART tokenizer is more likely to produce inconsistent tokenization (Table 1), and can therefore provide us with more exemplary results. For each dataset, we fine-tune two variants of the model: the first variant (denoted as `original`) tokenizes gold answers separately with the contexts, and the second variant (denoted as `consistent`) applies our method to guarantee consistent tokenization. Appendix C includes details of hyperparameters tuning and computing resources.

## 5 Findings

We run model training over each of the SQuAD, TriviaQA and NewsQA training sets, and report evaluation results on their corresponding dev sets in Table 2 ($F_1$ scores) and Appendix D (EM scores).

**Consistent tokenization training improves in-domain QA performance.** Overall, we observe statistically significant improvement in $F_1$ with the `consistent` variants on all of SQuAD, TriviaQA and NewsQA, with in-domain performance gains of 1.0, 1.1 and 0.6 for the three datasets, respectively (marked by shaded cells). A similar observation is made with the EM results. One potential explanation for the improvement is that, the task becomes inherently simpler with consistent tokenization. We hypothesize that consistent tokenization allows the model to extract answers from the context instead of also needing to learn to paraphrase the answer into something that does not exist in the context, as we mentioned in Section 3. We will validate this hypothesis when we examine the convergence speed of the models with or without consistent tokenization during training.

**Consistent tokenization also improves zero-shot QA performance on out-of-domain datasets.** We also examine zero-shot model performance on unseen domains in Table 2. We find that on all the OOD datasets, the corresponding $F_1$ of the

| Training Set | Tokenization | SQuAD | NewsQA | NQ | SearchQA | TriviaQA | BioASQ | DuoRC | TextbookQA | Average |
|---|---|---|---|---|---|---|---|---|---|---|
| **SQuAD** | `original` | 87.5 | 53.5 | 48.7 | 23.4 | 60.1 | 54.4 | **53.2**$^*$ | 41.0 | 47.8 |
| | `consistent` | **88.5**$^*$ | **55.0**$^*$ | **50.2**$^*$ | **25.7**$^*$ | **61.2**$^*$ | **58.5**$^*$ | 52.8 | **43.1**$^*$ | **49.5**$^*$ |
| **TriviaQA** | `original` | 54.6 | 35.5 | 44.6 | 62.8 | 75.5 | 35.7 | 41.0 | 37.2 | 44.5 |
| | `consistent` | **58.7**$^*$ | **39.9**$^*$ | **47.3**$^*$ | **63.6**$^*$ | **76.6**$^*$ | **40.4**$^*$ | **45.6**$^*$ | **39.3**$^*$ | **47.8**$^*$ |
| **NewsQA** | `original` | 75.8 | 65.0 | 55.5 | **38.4**$^*$ | **63.8**$^*$ | 50.7 | 56.5 | 43.4 | 56.2 |
| | `consistent` | **77.7**$^*$ | **65.6**$^*$ | **56.5**$^*$ | 37.3 | 63.5 | **52.3**$^*$ | **57.1**$^*$ | **45.4**$^*$ | **56.9**$^*$ |

Table 2: F$_1$ of BART QA models fine-tuned on different datasets (first column) and evaluated on in-domain and out-of-domain datasets. `Original` represents models fine-tuned with original tokenization and `consistent` represents models fine-tuned with consistent tokenization (our method). Shaded cells indicate in-domain evaluation results. All results are averaged over three random seeds. * marks results with statistically significant improvement ($p < 0.05$) over the other model variant on the same dataset.

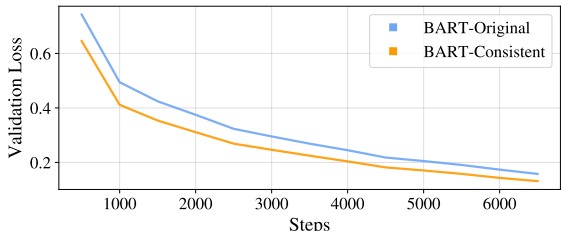

(a) Learning Curve

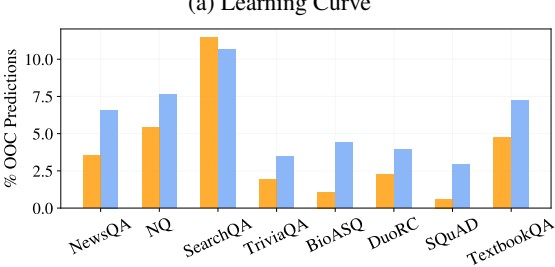

(b) Percentage of Out-of-Context Prediction

Figure 2: (a) Learning curve of BART with original tokenization and consistent tokenization. (b) Percentage of instances that model generates out-of-context answers during inference. The models are trained on SQuAD and numbers are averaged over three random seeds. Appendix G reports percentage of out-of-context answers when the models are trained on TriviaQA and NewsQA.

`consistent` model is either comparable or higher than the `original` model, with the highest gain more than 4%. This implies that training with consistent tokenization systematically improves model generalization beyond better overfitting the training domain. The improvement on unseen domain can also be explained by the reduction of difficulty: training with consistent tokenization provides the model with information that the answer must exist within the input, thus the model is no longer required to extensively search the entire vocabulary on a previously unknown domain.

**Training with consistent tokenization leads to faster model convergence, and improved model confidence on the gold answer.** We present the training curves of models fine-tuned on SQuAD in Figure 2a, and include training curves for other datasets in Appendix E. Across all the datasets, the `consistent` models converge faster than the `original` ones. This corroborates our hypothesis that solving extractive QA by simply extracting answers at the token level can make the task easier.

To further validate this, we also examine the log perplexity of the gold answer for each instance, and compare the distributional difference between `consistent` and `original` models (shown in Figure 5, Appendix F). We see that the overall distribution of log perplexity difference leans to the negative side, suggesting that the model is more confident in generating gold answers when tokenization consistency is enforced during training.

**Training with consistent tokenization leads to less textual hallucination.** An important angle to look at generative models is hallucination. It is worth noting that while the general meaning of hallucination, or more precisely *factual hallucination*, refers to the problem when a model produces factual content that is either inconsistent with or ungrounded to the input. In the context of extractive QA, we instead refer to a more specific definition of *textual hallucination*, where the QA model produces *out-of-context* answer sequence that does not align with any given span of text in the input. We show the percentage of instances that models generate out-of-context answers on different datasets in Figure 2b. An example of such out-of-context answers can be found in Appendix H.

We find that when trained with consistent tokenization, the model is less likely to textually hallucinate. We conjecture that this is because with inconsistently tokenized training data, the model is undesirably exposed to a many context-answer pairs that are not directly aligned at the token level.

This misalignment between the tokenized answer and its inconsistent form in the tokenized context possibly leads to the higher hallucination rate.

# 6 Conclusion

We identify and address the issue of tokenization consistency in extractive tasks. We find that consistent tokenization improves model training on in-domain performance, convergence speed, out-of-domain generalization, and textual hallucination. It is worth noting that inconsistent tokenization may affect any extractive tasks. Using these findings, we suggest to apply consistent tokenization to inputs and outputs whenever researchers or practitioners are tackling extractive tasks.

# 7 Limitations

In this work, we only investigate consistent tokenization in fine-tuning. Future work might consider focusing on the effect of consistent tokenization in in-context learning. In addition, it is possible that fine-tuning with consistent tokenization does not align with model's pre-training objective, but the role of pre-training objective is also not explored in this work. Furthermore, besides BPE tokenizers, there are also other tokenizers that do not produce encoding by merging subwords. Whether these tokenizers will produce a non-negligible amount of inconsistent tokenizations is unknown. Finally, the datasets used for experiments are solely English question answering datasets. Whether the same observation holds in multilingual setting or in other tasks requires further examination.

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

# A Percentage of Inconsistently Tokenized Samples After Adding Prefix Space

| Dataset | BART | T5 | Dataset | BART | T5 |
|---------|------|------|-----------|------|------|
| SQuAD   | 3.9  | 5.0  | BioASQ    | 25.1 | 25.3 |
| TriviaQA| 72.7 | 72.8 | TextbookQA| 22.0 | 21.8 |
| NewsQA  | 7.8  | 0.0  | DuoRC     | 27.8 | 27.4 |
| NQ      | 0.1  | 0.4  | SearchQA  | 17.0 | 14.7 |

Table 3: Percentage of training instances whose tokenized gold answers do not exist verbatim in the tokenized context using BART and T5 tokenizer, when the inconsistency caused by prefix space is addressed.

One of the most well-known source of tokenization inconsistency is the prefix space, which may not be contained in the output but might be contained in the appearance of output sequence in the input. [Explain with a tangible example, and spell out the implication once again: "which may not.." → where the output is usually not prefixed with whitespace (e.g., "1912" in Figure 1), but its appearance in the input usually is ("... in 1912"). When the tokenizer (e.g., BPE) is sensitive to such minute differences, the tokenization of the output sequence can differ drastically for the output alone compared to its appearance in the input (["19", "12"] vs ["Ġ1912"]).]$_{Peng}$ This issue can be easily resolved by adding a prefix space during tokenization of the output, and we show a percentage of inconsistently tokenized instances in Table 3 when prefix space is resolved. [Explain that a lot of issues cannot be addressed with prefix space – be consistent with the messaging in the main text]$_{Peng}$

# B License of Artifacts and Intended Use

Table 4 include a list of artifacts used in this work and their intended usage. Our use of these artifacts aligns with their intended usage.

| Artifact | License | Intended Usage |
|----------|---------|----------------|
| MRQA | MIT | A benchmark focuses on generalization in extractive QA format |
| BART | Apache-2.0 | A pre-trained model for sequence-to-sequence tasks. |

Table 4: License and intended usage for the artifacts we used.

Dataset statistics for each dataset we used is included in Table 5.

# C Hyperparameters

Before fine-tuning, we conduct minimal hyperparameter search across learning rate of $1 \times 10^{-5}$,

| Dataset | Training | Dev/Test |
|---------|----------|----------|
| SQuAD | 86588 | 10507 |
| NaturalQuestions | 104071 | 12836 |
| NewsQA | 74160 | 4212 |
| TriviaQA | 61688 | 7785 |
| SearchQA | 117384 | 16980 |
| BioASQ | - | 1504 |
| TextbookQA | - | 1503 |
| DuoRC | - | 1501 |

Table 5: Number of instances for each dataset.

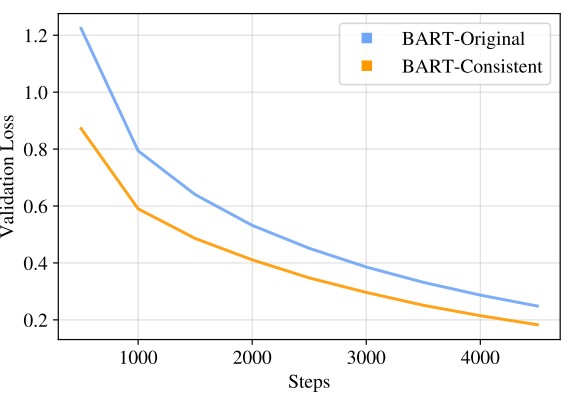

Figure 3: Learning curve of BART with original tokenization and consistent tokenization. The models are trained on TriviaQA.

$2 \times 10^{-5}$, and $3 \times 10^{-5}$.

For each training set, we fine-tune the model with three different random seeds and average the resulting metrics for final performance. For each fine-tuning trial, we used 4 Nvidia V100 GPUs, each of which has 16GB CUDA memory.

We use a learning rate of 2e-5 and effective batch size of 128. The maximum sequence length is set to 1024. The model is trained for 10 epochs and the best checkpoint is selected based on the performance of in-domain development set.

The fine-tuning code is based on Huggingface Transformers [2]. The implementation details can be found in our released repository.

# D In-domain and Out-of-domain Exact Match Accuracy

The exact match accuracy is shown in Table 6. Similar to what we discover in section 5, the model obtains a better performance whent training with consistent tokenization.

| Training Set | Tokenization | SQuAD | NewsQA | NQ | SearchQA | TriviaQA | BioASQ | DuoRC | TextbookQA | Average |
|---|---|---|---|---|---|---|---|---|---|---|
| **SQuAD** | original | 79.2 | 37.4 | 33.6 | 16.5 | 52.2 | 42.1 | 43.8 | 29.8 | 36.5 |
|  | consistent | **80.4** | **38.4** | **33.9** | **19.3** | **54.0** | **46.6** | **44.2** | **32.0** | **38.4** |
| **TriviaQA** | original | 43.5 | 22.9 | 30.8 | 54.0 | 70.7 | 26.2 | 32.2 | 30.7 | 34.3 |
|  | consistent | **47.8** | **26.9** | **33.3** | **55.5** | **72.2** | **30.4** | **35.8** | **32.6** | **37.5** |
| **NewsQA** | original | 61.8 | 49.2 | 41.0 | **31.5** | 55.9 | 35.5 | 45.3 | 33.1 | 44.2 |
|  | consistent | **64.8** | **49.7** | **42.3** | 30.6 | **56.0** | **37.3** | **45.7** | **34.7** | **45.1** |

Table 6: EM of BART fine-tuned on different datasets (first column) and evaluated on in-domain and out-of-domain datasets. `Original` represents models fine-tuned with original tokenization and `consistent` represents models fine-tuned with consistent tokenization. All results are averaged over three random seeds.

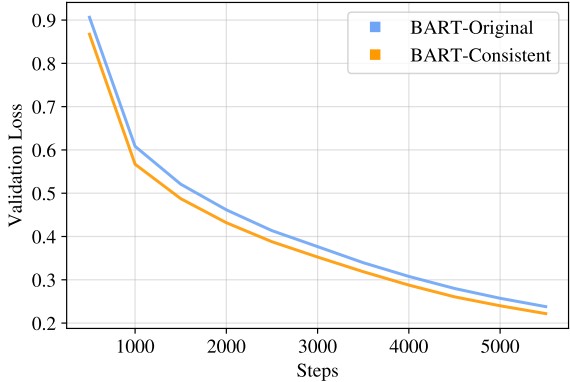

Figure 4: Learning curve of BART with original tokenization and consistent tokenization. The models are trained on NewsQA.

## E   Learning Curve on Other Training Sets

The learning curves with models training on TriviaQA and NewsQA are shown in Figure 3 and 4, in both of which `consistent` model exhibits a faster convergence speed than that of `original` model.

## F   Log Perplexity Difference

Figure 5 shows the log perplexity difference between `consistent` and `original` models on in-domain dataset. We present an example of log perplexity difference on out-of-domain dataset in Figure 6, using BioASQ as an example.

## G   Textual Hallucination Rate on Other Training Sets

Figure 8 and 7 show textual hallucination rate when the models are trained on NewsQA and TriviaQA.

## H   Example of Out-of-Context Generation

Table 7 presents an example of out-of-context answer generated by the model.

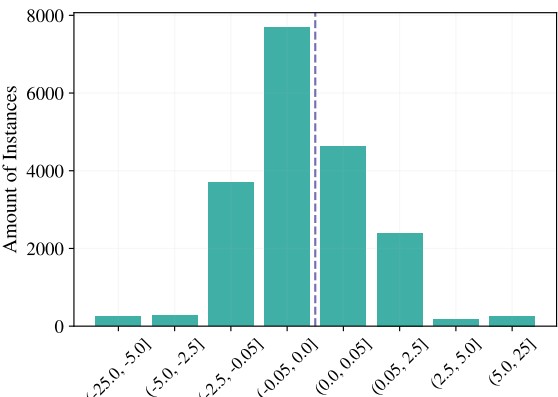

Figure 5: Instance amount distribution with respect to LP(`consistent`) - LP(`original`), where LP represents *log perplexity*. Model is trained on SQuAD. When the instance is located on the left of the dotted line (LP difference less than zero), the `consistent` model is more confident in generating gold answer than the `original` model.

| | |
|---|---|
| **Context:** | CBS set the base rate for a 30-second advertisement at $5,000,000, a record high price for a Super Bowl ad. As of January 26, the advertisements had not yet sold out. CBS mandated that all advertisers purchase a package covering time on both the television and digital broadcasts of the game, meaning that for the first time, digital streams of the game would carry all national advertising in pattern with the television broadcast. This would be the final year in a multi-year contract with Anheuser-Busch InBev that allowed the beer manufacturer to air multiple advertisements during the game at a steep discount. It was also the final year that **Doritos**, a longtime sponsor of the game, held its "Crash the Super Bowl" contest that allowed viewers to create their own Doritos ads for a chance to have it aired during the game. Nintendo and The Pokémon Company also made their Super Bowl debut, promoting the 20th anniversary of the Pokémon video game and media franchise. |
| **Question:** | Which company has held contests for fans to create their own ad for the company? |
| **Gold Answer:** | Doritos |
| **Prediction:** | Dorfos |
| **Context:** | LONDON, England (CNN) – **Israeli military action in Gaza is comparable to that of German soldiers during the Holocaust**, a Jewish UK lawmaker whose family suffered at the hands of the Nazis has claimed. A protester confronts police in London last weekend at a demonstration against Israeli action in Gaza. Gerald Kaufman, a member of the UKś ruling Labour Party, also called for an arms embargo on Israel, currently fighting militant Palestinian group Hamas, during the ... |
| **Question:** | What does the lawmaker say? |
| **Answer:** | Israeli military action in Gaza is comparable to that of German soldiers during the Holocaust |
| **Prediction:** | Nazi soldiers during the Holocaust |

Table 7: Examples of out-of-context predictions made by model. In the first example from SQuAD, the model is spelling the answer incorrectly; and the second example the model outputs an incorrect (non-extractive) answer although it is also factually incorrect.

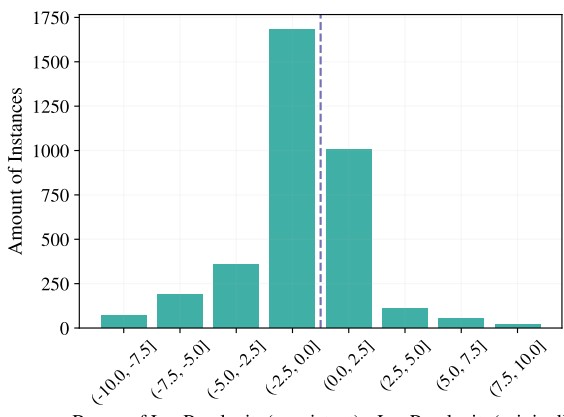

Figure 6: Instance amount distribution with respect to LP(`consistent`) - LP(`original`) in BioASQ development set, where LP is log perplexity. Model is trained on SQuAD. When the instance is located on the left of the dotted line (LP difference less than zero), the `consistent` model is more confident in generating gold answer than the `original` model. Compare to Figure 5, this figure shows an example of log perplexity difference on out-of-domain datasets.

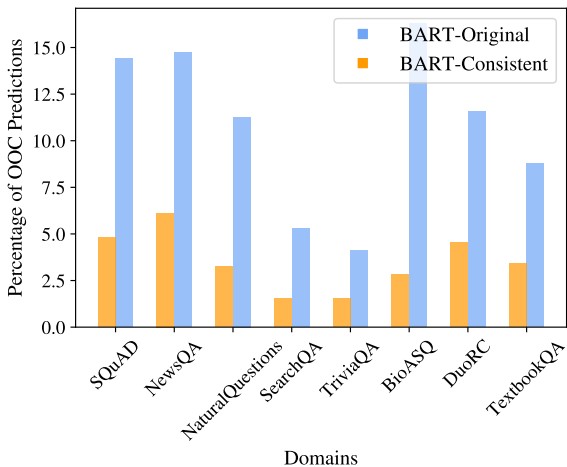

Figure 7: Percentage of instances that model generates out-of-context answers during inference. The models are trained on TriviaQA and numbers are averaged over three random seeds.

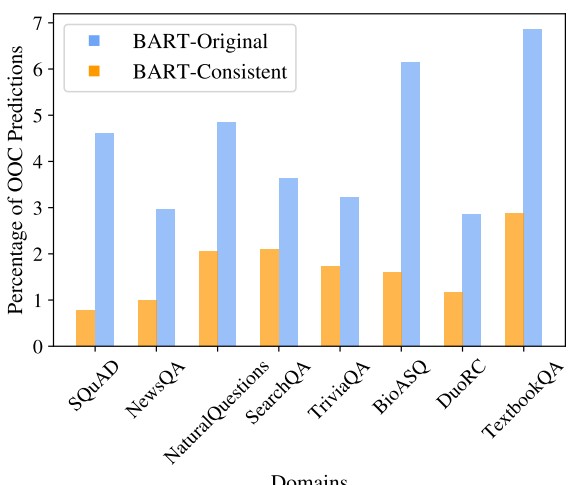

Figure 8: Percentage of instances that model generates out-of-context answers during inference. The models are trained on NewsQA and numbers are averaged over three random seeds.