# OpenReview forum: "Tokenization Consistency Matters for Generative Models on Extractive NLP Tasks"
_EMNLP/2023/Conference — EMNLP 2023 Findings_

### Official Review · Reviewer_rr2n · 2023-08-02

**Soundness:** 3

**Excitement:**

4: Strong: This paper deepens the understanding of some phenomenon or lowers the barriers to an existing research direction.

**Paper Topic And Main Contributions:**

The paper highlights an important issue often overlooked in training generative models for extractive tasks - tokenization inconsistency. The authors demonstrate how this inconsistency can hinder the performance of these models, leading to a drop in effectiveness as well as the generation of incorrect information. To address this issue, the authors propose a simple yet effective fix and present a case study focusing on extractive question answering (QA). The results indicate that with consistent tokenization, the model performs better on both in-domain and out-of-domain datasets when evaluated on various QA datasets. Additionally, the model exhibits faster convergence and a reduced likelihood of generating out-of-context answers. This paper serves as a reminder of the importance of scrutinizing tokenization in extractive tasks and highlights the benefits of consistent tokenization during training.


**Reasons To Accept:**

The paper brings attention to an often-neglected issue in the field of generative models for extractive tasks, shedding light on the impact of tokenization inconsistency.

By focusing on extractive QA, the authors provide a concrete example to illustrate the effects of consistent tokenization, making their findings more relatable and applicable.

The results of the case study demonstrate a significant improvement in model performance, with an average F1 gain of +1.7, enhancing the credibility and value of the proposed solution.

The authors highlight that consistent tokenization leads to faster convergence, which is a valuable advantage when training generative models for extractive tasks.

The paper addresses the issue of out-of-context answers, further emphasizing the importance of consistent tokenization and its impact on generating accurate and coherent responses.


**Reasons To Reject:**

The paper primarily focuses on extractive QA, which may limit the generalizability of the proposed solution to other extractive tasks. Expanding the scope to include a broader range of tasks could strengthen the overall impact of the research. The paper utilizes eight QA datasets for evaluation, which may not encompass the full diversity of extractive tasks. Including a more diverse range of datasets could enhance the validity and applicability of the results.

While the paper demonstrates the effectiveness of the proposed fix, a direct comparison with other existing methods for addressing tokenization inconsistency would provide a more comprehensive evaluation.



**Reproducibility:**

3: Could reproduce the results with some difficulty. The settings of parameters are underspecified or subjectively determined; the training/evaluation data are not widely available.

**Reviewer Confidence:**

4: Quite sure. I tried to check the important points carefully. It's unlikely, though conceivable, that I missed something that should affect my ratings.

---

> ### Author Rebuttal · Authors · 2023-08-29
>
> Thank you for your careful review. Below is our response:
>
> _“The paper primarily focuses on extractive QA, which may limit the generalizability of the proposed solution to other extractive tasks.”_
>
> * We agree that increasing the amount of tasks used for the case study can benefit the analysis. In the meantime, most of the analysis is on the problems that matter for all extractive NLP tasks, and we observe consistent trends across models trained on different datasets, we are moderately confident that the same observation can be seen on other tasks.
> * Since our work primarily intends to identify the problem and perform an in-depth investigation of how many issues can be brought by tokenization consistency, extending the work to other tasks and datasets (other than the eight we used) should enlarge the scope of our work, thus making it unsuitable for a short paper.
>
> _“A direct comparison with other existing methods for addressing tokenization inconsistency”_
>
> * Thank you for your suggestion! We want to clarify that because the problems identified in prior works (Petrak et al., Rosenbaum et al. in Sec 2) are only a subset of the consistency issue we discuss here, and the main focus of those works is not on tokenization of arbitrary text, to our knowledge, there are no methods that address the same problem as we describe.
> * We are more than happy to include a more detailed description of the methods described in prior works to explain how the problem and solutions are different.

---

### Official Review · Reviewer_Yyrb · 2023-08-10

**Soundness:** 4

**Excitement:**

3: Ambivalent: It has merits (e.g., it reports state-of-the-art results, the idea is nice), but there are key weaknesses (e.g., it describes incremental work), and it can significantly benefit from another round of revision. However, I won't object to accepting it if my co-reviewers champion it.

**Paper Topic And Main Contributions:**

Summary: The paper focuses on the use of generative models in extractive tasks, such as extractive question answering. The primary issue highlighted is the problem of tokenization inconsistency which is often overlooked during training of BART. This inconsistency in how the input and output are tokenized affects the extractive nature of tasks, leading to performance degradation and instances of hallucination.

Main Contributions:
1. Identification of the Problem: The paper points out the tokenization inconsistency in the context of extractive tasks, which is a contribution to understanding the pitfalls in generative model training for such tasks.
2. Sufficient experiments on extractive QA task: By simply aligning the tokenization in the original text and ground truth, the authors conduct experiments on several popular extractive QA datasets, and observe consistent improvements.

**Questions For The Authors:**

My main concern is the first point in weakness, i.e., is this tokenization inconsistency specific to BART? Does this problem exist in other models as well? And does this inconsistency occur in a high percentage of actual datasets?

**Reasons To Accept:**

1. The authors highlight a subtle yet significant inconsistency in tokenization within the BPE of the BART model, suggesting this might be a latent factor hindering its optimal performance in extractive tasks.
2. By conducting thorough experiments, the paper showcases that aligning the tokenization of input text with the ground truth can enhance the BART model's efficacy in extractive QA tasks.

**Reasons To Reject:**

1. Insufficient evidence that the problems identified in this paper are widespread and important. For example, what is the approximate percentage of such tokenization inconsistencies in the actual data? Do other models, such as the T5, GPT series, also have this problem with their BPE?
2. The problem identified in this paper is primarily a BPE problem. Therefore the proposed method is more like trick than an essential solution to the problem.

**Reproducibility:**

4: Could mostly reproduce the results, but there may be some variation because of sample variance or minor variations in their interpretation of the protocol or method.

**Reviewer Confidence:**

3: Pretty sure, but there's a chance I missed something. Although I have a good feel for this area in general, I did not carefully check the paper's details, e.g., the math, experimental design, or novelty.

---

> ### Author Rebuttal · Authors · 2023-08-29
>
> Thank you for your careful comments. Below is our response:
>
> _“Insufficient evidence that the problems identified in this paper are widespread and important.”_
>
> * We want to note that similar problems are found in some prior works we mentioned in section 2  (L94-99), while the problems identified in these prior works are subsets of the inconsistency issue we describe here and the main focus of their works was not tokenization. Geva and Gupta et al. (2022, https://aclanthology.org/2020.acl-main.89/) and Petrak et al. (2022, https://arxiv.org/abs/2205.06733)  pointed out the issue of tokenizing numerical values, which is a subset of the inconsistency problem that we study. The same inconsistency in semantic parsing is also mentioned by Rosenbaum et al. (2022, https://aclanthology.org/2022.aacl-short.56/).
> * Table 1 and 3 also provides statistics on the amount of inconsistently tokenized instances in the eight datasets. For these commonly used QA datasets, all of them have a nonignorable amount of instances suffering from this issue.
> * Additionally, this problem may be easily disregarded in the training process, while fixing it can bring improvement on performance, generalization, hallucination, and convergence speed. We hope the results we present can raise some attention to the existence of this problem.
>
> _“What is the approximate percentage of such tokenization inconsistencies in the actual data?”_
>
> * We believe that the definition of “actual data” matters for the approximate percentage here. In this paper, We use the widely adopted extractive QA as a case study to show the existence of this issue.
> * For other tasks, as we mentioned in the previous bullet point, multiple prior works have identified a subset of the issue we are discussing, while the main focus of their work is not on tokenization.
> * We’d like to clarify that we hope the results we present can raise some attention to the existence of this problem thereby benefiting the community by producing higher-quality models. Although it would be valuable and practical, we did not intend to conduct comprehensive experiments on real-world data, as this would be out of the scope of a short paper.
>
> _“The problem identified in this paper is primarily a BPE problem. Therefore the proposed method is more like trick than an essential solution to the problem.”_
>
> * Given the common use of BPE tokenizer and its variants, we believe that tokenization inconsistency can be an important issue and costly to overlook.
> * We agree that it is a better option to propose a tokenization method that is inherently immune to tokenization inconsistency. However, the method we proposed is guaranteed to solve the issue and does not require much change to the training process. Therefore it is a plug-and-play method for any model that uses BPE tokenizer or its variants. We believe that the simplicity of this solution can help more researchers mitigate this issue in future research without any effort.
>
> _“Do other models, such as the T5, GPT series, also have this problem with their BPE?”_
>
> * We present statistics of affected instances on T5’s SentencePiece tokenizer in Table 1 and 3, in which SentencePiece is less prone to tokenization inconsistency (as it incorporates a unigram tokenizer as an additional mechanism), but still show a non-ignorable amount of instances that have the issue.

---

### Official Review · Reviewer_gJMX · 2023-08-11

**Soundness:** 4

**Excitement:**

3: Ambivalent: It has merits (e.g., it reports state-of-the-art results, the idea is nice), but there are key weaknesses (e.g., it describes incremental work), and it can significantly benefit from another round of revision. However, I won't object to accepting it if my co-reviewers champion it.

**Paper Topic And Main Contributions:**

This paper shows that, with simple hot-patching BPE tokenization strategy which ensures consistent tokenization over generative models, extractive paradigms with these models can perform better in various QA datasets with notable metric improvement.

The main contribution of this paper is a simple yet effective approach with extractive QA as a testbed, where consistent tokenized answers rather than paraphrased answers are used to directly supervise generation.

**Questions For The Authors:**

A. I am wondering if the authors have tried to validate the same idea on sentencepiece tokenizer. While L131 mentions `less subject to inconsistency`, such results will still benefit the community, since copying from context is a wide-use scenario of current generative foundation models (with sentencepiece tokenizers).

**Reasons To Accept:**

S1. the proposed method is easy to understand and straightforward with consistent metric improvement on most datasets.

S2. the author spotted an exciting fact in generative machine reading comprehension (MRC) which might attract future investigation.

S3. evaluation is basically fair and comprehensive, demonstrating improvements in both in-domain and out-domain settings.

**Reasons To Reject:**

W1. Although the proposed tokenization consistency issue is evident in the extractive QA setting through the experiments, I strongly suspect whether it is possible that this inconsistency issue can be easily dissipated by the model and data scale.

W2. Current conclusions are confined to the BPE tokenizer, limiting the feasibility of the proposed approach in models equipped with other tokenizers.

**Reproducibility:**

5: Could easily reproduce the results.

**Reviewer Confidence:**

4: Quite sure. I tried to check the important points carefully. It's unlikely, though conceivable, that I missed something that should affect my ratings.

---

> ### Author Rebuttal · Authors · 2023-08-29
>
> Thank you for your detailed comments. Below is our response:
>
> _“W1. Whether it is possible that this inconsistency issue can be easily dissipated by the model and data scale.”_
>
> * It is a very intriguing question regarding whether tokenization inconsistency can be resolved by having (a) more fine-tuning data, (b) larger models, or (c) more pretraining.
> * However, none of the three potential solutions mentioned above can fundamentally resolve the issue of tokenization inconsistency. Even larger models with tons of training/pretraining data observe different inconsistently tokenized examples at different frequencies, which limits their generalization capabilities to unseen/less seen examples. Instead of relying on expensive and uncertain parametric learning of this behavior, we offer a simple yet principled solution to the problem.
>
> _“Q(a). & W2  If the authors have tried to validate the same idea on SentencePiece tokenizer. While L131 mentions less subject to inconsistency”_
>
> * We have run some initial experiments with SentencePiece tokenizer on T5v1.1 with a single random seed. As we have mentioned in Tables 1 and 3, the number of instances in the eight datasets we chose subject not as much the inconsistency issue, and our initial results on T5 only showed modest improvement (In-domain EM improvement $\leq 0.5$%, without significance test as only single random seed is run). In the meantime, we are happy to include these initial results in the appendix in case it may be helpful to the community.
> * In principle, the same effect applies to SentencePiece and other tokenizers. For further experiments on SentencePiece, we believe that it would be beneficial to manually craft some datasets that SentencePiece struggles with producing consistent tokenizations. However, this is out of the scope of our current (short) paper.

---

### Official Review · Reviewer_ty6n · 2023-08-12

**Soundness:** 4

**Excitement:**

4: Strong: This paper deepens the understanding of some phenomenon or lowers the barriers to an existing research direction.

**Paper Topic And Main Contributions:**

This paper highlights an important issue with span based question answeing tasks, mainly how incorrect tokenization can lead to degradation in performance.

**Reasons To Accept:**

1) The methods provided are concise and clear.
2) The results justify the initial assumptions and are exhaustive.

**Reasons To Reject:**

1) This sort of work merits an extensive error analysis in terms of when this method is likely to be most efficient and situations in which it might not be as efficient.

**Reproducibility:**

4: Could mostly reproduce the results, but there may be some variation because of sample variance or minor variations in their interpretation of the protocol or method.

**Reviewer Confidence:**

4: Quite sure. I tried to check the important points carefully. It's unlikely, though conceivable, that I missed something that should affect my ratings.

---

> ### Author Rebuttal · Authors · 2023-08-29
>
> Thank you for your thoughtful comments. We appreciate your feedback.

---

### Meta-Review · Senior_Area_Chairs · 2023-10-05

**Recommendation:** 4

**Metareview:**

This paper highlights the issue of inconsistent tokenization between the input and output sequences in extractive QA with BPE tokenization (e.g. as done in the BART pretrained model). The authors then suggest a method to ensure consistency between the input and output which improves performance across several datasets. The highlighted problem is important to be aware of as it can be found in many extractive tasks where generative models are applied and the suggested method seems to be a good remedy (although not very exciting). The suggested experiments and results are sound. Given the above, this paper may be a good fit fir either Findings or the main conference.

---

### Decision · Program_Chairs · 2023-10-07

**Decision:**

Accept-Findings

**Comment:**

This paper highlights the issue of inconsistent tokenization between the input and output sequences in extractive QA with BPE tokenization (e.g. as done in the BART pretrained model). The authors then suggest a method to ensure consistency between the input and output which improves performance across several datasets. The highlighted problem is important to be aware of as it can be found in many extractive tasks where generative models are applied and the suggested method seems to be a good remedy (although not very exciting). The suggested experiments and results are sound. Given the above, this paper may be a good fit fir either Findings or the main conference.